



# Development of a comprehensive data basis of scattering environmental conditions and simulation constraints for offshore wind turbines

Clemens Hübler[1], Cristian Guillermo Gebhardt[1], and Raimund Rolfes[1]

[1]Institute of Structural Analysis, Leibniz Universität Hannover, Appelstr. 9a, D-30167 Hannover, Germany

*Correspondence to:* Clemens Hübler (c.huebler@isd.uni-hannover.de)

**Abstract.** For the design and optimisation of offshore wind turbines, the knowledge of realistic environmental conditions and utilisation of well-founded simulation constraints is very important, as both influence the structural behaviour and power output in numerical simulations. However, real high-quality data, especially for research purposes, is scarcely available. This is why, in this work, a comprehensive data basis of thirteen environmental conditions at wind turbine locations in the North and Baltic Sea is derived using data of the FINO research platforms. For simulation constraints, like the simulation length and start-up time, well-founded recommendations in literature are also rare. Nevertheless, it is known that the choice of simulation lengths and start-up times fundamentally affects the quality and computing time of simulations. For this reason, studies of convergence for both parameters are conducted to determine adequate values depending on the type of substructure, the wind speed and the considered loading (fatigue or ultimate). As the main purpose of both the data basis and the simulation constraints is to compromise realistic data for probabilistic design approaches and to serve as a guidance for further studies in order to enable more realistic and accurate simulations, all results are freely available and easy to apply.

## 1 Introduction

Although the share of offshore wind energy in overall energy production has been steadily growing over the last years, the cost of offshore wind energy is still high compared to other renewable energies (Kost et al., 2013). In order to achieve potential cost reductions of about 30 % in the next ten years (Prognos AG and Fichtner, 2013), a realistic and accurate simulation of offshore wind turbines and their substructures is beneficial. On the one hand, for realistic simulations, the knowledge of scattering environmental conditions is a central point. On the other hand, carefully chosen simulation constraints, like the simulation length or start-up time, are essential to obtain accurate results.

Regarding the first point, current guidelines (IEC, 2009) already define that simulations should mirror the changing environmental conditions at the precise site of a wind turbine. However, for academic research, real site data is rarely available, and even for industrial purposes data quality might be poor for some parameters or long-term data is missing. As a result, various research projects characterised environmental conditions at specific sites or entire areas, and published statistical distributions as a reference. Probably the most frequently used example is the UPWIND design basis (Fischer et al., 2010). Further examples are the work of Stewart et al. (2015), the PSA-OWT project (Hansen et al., 2015), and the investigations by Häfele et



al. (2017). All these reference conditions have some limitations. The design basis of Stewart et al. (2015) is only for deep water sites off the coasts of the United States of America. Inter alia, the wave state of deep water sites are not comparable to shallow water conditions in the North Sea, as significant wave heights generally increase with the water depth (Hansen et al., 2015). Additionally, wind speeds are not measured at hub height, and therefore, have to be extrapolated, which increases

uncertainties. For the UPWIND design basis, the wind speed is just given at a reference height of 10 m and not at hub height as well. Furthermore, no statistical distributions for conditional parameters (e.g. the wave height $H_s$ depends on the wind speed $v_s$) are given, but only scatter plots. In the PSA-OWT project, data of the research platform FINO1 in the North Sea is used. Here, the wind speed is measured at hub height, but shadow effects can occur, if sensors are positioned behind the measuring mast. Häfele et al. use data of the research platform FINO3, which has several sensors at each height to reduce shadow effects.

However, only five environmental parameters (wind speed and direction, wave height, period and direction) are analysed, and the data period is only five years. Hence, the need for a comprehensive data basis, covering several sites and the most important parameters, becomes obvious in order to enable future research that is based on realistic data. Missing conditions are for example the turbulence intensity, the wind shear or ocean currents.

As to the second point, simulation constraints are frequently chosen based on experience, literature values or recommendations

in current standards. However, considering the simulation length and start-up time, recommendations in the guidelines are mainly fairly vague (GL, 2012; IEC, 2009). Simulation lengths of 10 minutes for fatigue calculations (FLS), and one hour or less for ultimate loads (ULS) are frequently recommended, and start-up times of 5 seconds or more. Literature values partly differ significantly. For the start-up time, values of 20, 30 or 60 seconds are chosen for example (Vemula et al. (2010); Jonkman and Musial (2010); Hübler et al. (2017)), and simulation lengths of 10 minutes and one hour are common practice (Jonkman

and Musial, 2010; Popko et al., 2012; Cheng, 2002). However, longer simulation lengths are partly used as well, especially in the oil and gas industry or for floating substructures (DNV, 2013). Still, all these recommendations are not underpinned with detailed analyses. For floating offshore wind turbines, such investigations were conducted for the simulation length by Stewart et al. (2015), Stewart et al. (2013) and Haid et al. (2013). It is shown that simulation lengths of 10 minutes are sufficient for ULS and FLS loads. The observation that ULS and FLS loads tend to be higher for longer simulations are not due to physical

reasons, but to unclosed cycles in the Rainflow counting for the FLS case and a result of the averaging technique in case of ULS loads. Both can be handled by adapting the algorithms. Concerning the start-up time, Haid et al. (2013) recommend 60 seconds and the utilisation of initial conditions. This recommendation is based on an analysis which has not been further specified. For a jacket foundation, Zwick and Muskulus (2015) conducted a study investigating the simulation length and start-up time and also concluded that 10 minutes are sufficient, as long as 10-minute time series are merged before the Rainflow counting is applied.

The required start-up time is determined by checking the rotor speed to reach a steady state. However, neither initial conditions are applied, nor does a steady rotor speed guarantee that all transients are damped out. Therefore, the need for well-founded guidance on simulation lengths and start-up times for bottom fixed substructures becomes clear. For the simulation length, useful preliminary work is available, but it is limited to jacket substructures. Concerning the start-up time, extensive studies are rare, and do not concentrate on the convergence of the relevant loads (FLS and ULS). Furthermore, scattering environmental

conditions are not taken into account. This is a simplification especially in case of the start-up time, as this variation might lead





to more pronounced resonance effects (e.g. rarely occurring low wave peak periods that are close to the natural frequency of the structure; cf. Sect. 2.4) and therefore to higher start-up times.

After all, the listed shortcoming in state-of-the-art modelling assistance motivated the current work that focuses on the following aspects:

(1)  Derive an open access data basis for various scattering environmental conditions at different sites to enable more realistic modelling.

    (2)  Give well-founded guidance on simulation length and start-up time requirements, when these realistic conditions are applied, to improve accuracy of numerical simulations.

In order to address these topics, firstly, a data basis for all significant environmental conditions is derived from real data of the
FINO research platforms. In this work, the data source is introduced, the analysis is described, and the resulting distributions and some interesting findings are presented. Secondly, required simulation lengths and start-up times are determined. For this purpose, the simulation model is explained. Then, studies of convergence are conducted for the simulation length and start-up time. A monopile and a jacket substructure, FLS and ULS loads, and different wind speeds are considered. Recommendations are summarised. Lastly, the benefits and limitations of the current approach are summarised, and a conclusion is drawn.

## 2  Comprehensive data basis

### 2.1  Raw data

Environmental conditions can vary significantly among various turbine sites. As these states affect loads, and therefore, the design of offshore wind turbines, precise data of specific turbine location is valuable. Real site data is scarce, which is the reason for the formerly mentioned reference data bases (Fischer et al., 2010; Hansen et al., 2015; Stewart et al., 2015; Häfele et al.,
2017). These data bases define conditional, statistical distributions for some of the most important environmental conditions: Wind speed and direction, wave height, direction and peak period. However, other conditions are fixed for each wind speed or are set completely constant. The states of the frequently used UPWIND design basis are summarised in Table 1 as an example. In this study, scattering conditions are derived directly from offshore measurement data. The raw data is taken from the three FINO platforms, and conditional distributions for the following 13 environmental parameters are determined: wind speed and
direction, wave height, peak period and direction, turbulence intensity, wind shear exponent, speed and direction of the sub- and near-surface current, and air and water density. The FINO measurement masts are located in the North Sea and Baltic Sea, and are operated on behalf of the German Federal Ministry for the Environment, Nature Conservation, Building and Nuclear Safety (BMUB)[1]. The locations of the three FINO sites are marked in Fig. 1.

For all three sites, maximum, minimum, mean, and standard deviation values of the wind speed, measured at different heights
between 30 m and 100 m above mean sea level, are available for 10-minute intervals. Wind speeds are measured with cup and

---

[1]Raw data of the FINO platforms is freely available for research purposes. See www.fino-offshore.de/en/ for details.





**Table 1.** Environmental conditions (wind speed $v_s$, significant wave height $H_s$, wave peak period $T_p$, and turbulence intensity TI) of the K13 shallow water site (UPWIND design basis (Fischer et al., 2010)). The wind shear exponent is $\alpha = 0.14$, and wind and wave directions are usually set to zero, but scatter plots are available.

| $v_s$ (m s$^{-1}$) | 2 | 4 | 6 | 8 | 10 | 12 | 14 | 16 | 18 | 20 | 22 | 24 | 26 |
|---|---|---|---|---|---|---|---|---|---|---|---|---|---|
| TI (%) | 29.2 | 20.4 | 17.5 | 16.0 | 15.2 | 14.6 | 14.2 | 13.9 | 13.6 | 13.4 | 13.3 | 13.1 | 13.0 |
| $H_s$ (m) | 1.07 | 1.10 | 1.18 | 1.31 | 1.48 | 1.70 | 1.91 | 2.19 | 2.47 | 2.76 | 3.09 | 3.42 | 3.76 |
| $T_p$ (s) | 6.03 | 5.88 | 5.76 | 5.67 | 5.74 | 5.88 | 6.07 | 6.37 | 6.71 | 6.99 | 7.40 | 7.80 | 8.14 |

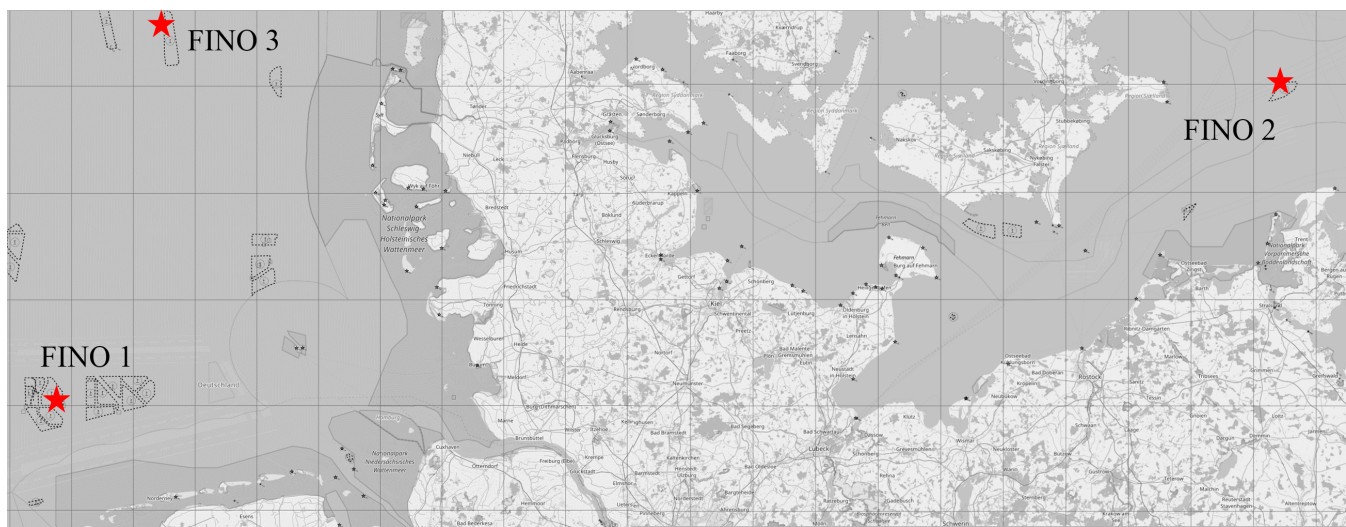

**Figure 1.** Positions of the three FINO platforms in the North and Baltic Sea, adapted from OpenStreetMap.

ultrasonic anemometers. In this study, cup anemometers are used, as these sensors are available at more different heights. For FINO1 and 2, the anemometers are positioned on jibs in secondary wind directions to reduce shadow effects. For FINO3, three anemometers are installed around the mast to minimise shadow effects. Sensors at different heights allow a detailed analysis of shear effects. Wind direction, air pressure, temperature and humidity are measured at different heights as well. Buoys in the immediate vicinity of the research platforms (about 150 m) measure the wave conditions. Mean values of significant wave heights, wave directions, wave peak periods and water temperatures are measured every 30 minutes. Furthermore, acoustic Doppler current profilers (ADCPs) close to the platforms measure ocean current velocities and directions at different water depths using the Doppler effect of sound waves. The platforms FINO1, 2 and 3 have been measuring continuously since 2004, 2007 and 2009 respectively, resulting in 7 to 13 complete years of measurement data, and enabling at least some long-term predictions. Data of incomplete years is not taken into account in order not to introduce bias due to seasonal effects.





## 2.2 Conditional distributions

In this work, raw data of the FINO measurement masts is used to set up a data base for correlated, scattering environmental conditions. As the post-processing of raw data is time-consuming and unnecessary to be repeated each time environmental conditions are used, conditional probability distributions (i.e. $P(Y = y | X = x)$ with $X$ being the independent random variable, $Y$ the dependent one, and $P$ the probability function) for environmental conditions are derived to make the data base easy to use. Firstly, post-processing is carried out to identify sensor failures (missing data) and measurement failures (outliers). Missing data is not interpolated, but left out, in order not to introduce any bias. As sufficient data of proper signal quality is available (e.g. more than $350\,000$ data points for the wind speed even for FINO3), this approach is practicable. Wind speed data is synchronised with the wind direction data. This enables a selection of the anemometer in front of the mast for FINO3. For FINO1 and 2, wind speed values are discarded, if the jib is located directly in the tower shadow. The turbulence intensity (TI) can be computed as the quotient of the standard deviation of the wind speed in a 10-minute interval ($\sigma_v$) and the mean wind speed in this interval ($v_s$) according to Eq. (1):

$$\mathrm{TI} = \frac{\sigma_v}{v_s}. \tag{1}$$

For the wind shear, Eq. (2) applies according to the standard IEC 61400-1 (2005):

$$v_s(z) = v_s(z_0) \times \left( \frac{z}{z_0} \right)^{\alpha}, \tag{2}$$

where $z$ is the height above mean sea level, $z_0$ is a reference height, $v_s(z)$ and $v_s(z_0)$ are wind speeds at the specified heights and $\alpha$ is the wind shear exponent. At the FINO platforms, the wind speed is measured at eight different heights. Therefore, it is possible to determine the wind shear exponent for every 10-minute interval by assuming $z_0 = 90\,\mathrm{m}$ and applying a non-linear regression. The air density can be calculated using Avogadro's Law in Eq. (3) and the measurements of humidity ($\phi$), air pressure ($p_\mathrm{humid}$), and temperature in degree Celsius ($T_\mathrm{air}$):

$$\rho_\mathrm{air} = \frac{p_\mathrm{humid}}{R_\mathrm{humid}\, T_\mathrm{air}}. \tag{3}$$

As humid air can be regarded as a mixture of ideal gases, the following equation applies for $R_\mathrm{humid}$:

$$R_\mathrm{humid} = \frac{R_\mathrm{dry}}{1 - \phi \frac{p_\mathrm{sat}}{p_\mathrm{humid}} \left( 1 - \frac{R_\mathrm{dry}}{R_\mathrm{vapour}} \right)}, \tag{4}$$

where $R_\mathrm{dry} = 287.1 \frac{\mathrm{J}}{\mathrm{kg\,K}}$ is the specific gas constant for dry air, $R_\mathrm{vapour} = 461.5 \frac{\mathrm{J}}{\mathrm{kg\,K}}$ for water vapour, and $p_\mathrm{sat}$ is the saturation vapour pressure that can, for example, be calculated using the August-Roche-Magnus formula:

$$p_\mathrm{sat} = 6.1094\,\mathrm{hPa} \times e^{\frac{17.625 \times T_\mathrm{air}}{T_\mathrm{air} + 243.04}}. \tag{5}$$

For the water density, a semi-analytical approach by Millero and Poisson (1981) of the following form is applied:

$$\rho_\mathrm{water} = A(T_\mathrm{water}) + B(T_\mathrm{water})S + C(T_\mathrm{water})S^{1.5} + DS^2, \tag{6}$$



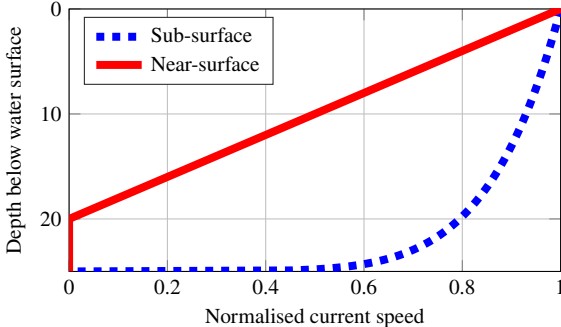

**Figure 2.** Velocity profiles of the sub- and near-surface currents according to Eqs. (7) and (8) respectively, with a water depth of 25 m and normalised speeds ($v_{\mathrm{SS},0} = v_{\mathrm{NS},0} = 1$)

where $S$ is the salinity, $T_{\mathrm{water}}$ is the water temperature at the surface, $A$, $B$ and $C$ are polynomial functions of the water temperature and $D$ is a constant. As constant salinity is assumed, the water density is a function of the water temperature. For all wave parameters, three-hour mean values are calculated, as wave conditions stay stationary for a duration of about three hours (GL, 2012). For the speeds and directions of sub- and near-surface currents, measured current values ($v_{\mathrm{m}}$ and $\theta_{\mathrm{m}}$) have

5 to be converted in order to separate sub- and near-surface components. According to, for example, IEC (2009), the following two equations apply for sub- and near-surface currents respectively:

$$v_{\mathrm{SS}}(z) = v_{\mathrm{SS}}(0\,\mathrm{m}) \left( \frac{d-z}{d} \right)^{\frac{1}{7}} \quad \text{and} \tag{7}$$

$$v_{\mathrm{NS}}(z) = \begin{cases} v_{\mathrm{NS}}(0\,\mathrm{m}) \left( \frac{20\,\mathrm{m}-z}{20\,\mathrm{m}} \right) & \text{for} \quad z <= 0 \\ 0 & \text{for} \quad z > 0. \end{cases} \tag{8}$$

Here, $v_{\mathrm{SS}}(z)$ and $v_{\mathrm{NS}}(z)$ are the sub- and near-surface current speeds at a position $z$ below the water surface, and $d$ is the water

10 depth. For reasons of clarity, the following notation is introduced: $v_{\mathrm{SS}}(z) = v_{\mathrm{SS},z}$. The velocity profiles are shown in Fig. 2. Obviously, the near-surface current does not exist below a reference depth of 20 m. Hence, it is possible to use measurement data of a depth of 20 m (or more) to directly get the sub-surface direction ($\theta_{\mathrm{SS},20} = \theta_{\mathrm{m},20}$) and to calculate the speed, for example for FINO2 ($d = 25\,m$):

$$v_{\mathrm{SS},0} = v_{\mathrm{SS},20} \left( \frac{25\,\mathrm{m}-20\,\mathrm{m}}{25\,\mathrm{m}} \right)^{-\frac{1}{7}}. \tag{9}$$

15 For the near-surface current, measurements close to the surface (e.g. $v_{\mathrm{m},2}$) can be used. However, these measurements include sub- and near-surface components, as shown in Fig. 3. Therefore, the sub-surface component at 2 m has to be calculated using Eq. (7), and the sub-surface direction is assumed to be constant over depth ($\theta_{\mathrm{SS},20} = \theta_{\mathrm{SS},2} = \theta_{\mathrm{SS},0}$). Then, trigonometrical





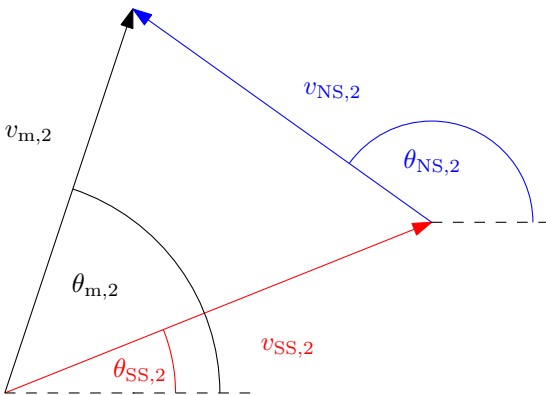

**Figure 3.** Vectorial analysis of ocean current components at a depth of 2 m (measured values (m), near- and sub-surface components (NS and SS))

relationships can be applied to calculate the near-surface current at 2 m:

$$v_{NS,2} = \sqrt{v_{SS,2}^2 + v_{m,2}^2 - 2v_{SS,2}v_{m,2}\cos(\theta_{m,2} - \theta_{SS,2})} \tag{10}$$

$$\theta_{NS,2} = \theta_{m,2} + \arcsin\left(v_{SS,2}\frac{\sin(\theta_{m,2} - \theta_{SS,2})}{v_{NS,2}}\right) \tag{11}$$

Lastly, the reference near-surface current $v_{NS,0}$ is given by:

$$v_{NS,0} = v_{NS,2}\left(\frac{20\,\text{m}}{20\,\text{m} - 2\,\text{m}}\right) \tag{12}$$

A depth-independent near-surface direction is assumed, and therefore, $\theta_{NS,0} = \theta_{NS,2}$.

After having post-processed the measurement raw data, maximum likelihood estimations are applied to the processed data of the regarded 13 environmental conditions in order to fit several statistical distributions. In addition to unimodal distributions, and if several distinct peaks are distinguishable, multimodal distributions are fitted as well, as it is assumed that the peaks are due to physical phenomena. However, as multimodal approaches have more degrees of freedom, they always fit the data better, even in case of a physically unimodal shape. Therefore, they have to be chosen with care in order not to fit physically unimodal distributions with multimodal approaches.

Considering the example of wind speed and wave height, it is self-evident that some environmental parameters are conditioned by others, and dependencies have to be defined. For example, the case of a calm sea during a storm is very unlikely. Analysing scatter plots of the environmental inputs and taking a literature review into account, the dependencies in Table 2 are defined, although it is possible to define them differently (cf. Stewart (2016)), as mainly the correlation is significant, and the determination of cause and effect is secondary.

One of the most common ways to include dependencies in statistical distributions is to split up the data of the dependent parameters into several bins of the independent parameters (e.g. Stewart (2016); Johannessen et al. (2002); Li et al. (2015)). To illustrate this approach, for example, the wave peak period is fitted in several bins of 0.5 m wave height (e.g.





**Table 2.** Dependencies, statistical distributions, and bin widths for environmental conditions derived from FINO1-3 data.

| Parameter | Statistical distributions | Dependencies | Bin sizes |
|---|---|---|---|
| Wind speed ($v_s$) | Weibull | – | – |
| Wind direction ($\theta_{\mathrm{wind}}$) | Non-parametric KDE | Wind speed | $2\,\mathrm{m\,s^{-1}}$ |
| Turbulence intensity (TI) | Weibull, gamma | Wind speed | $2\,\mathrm{m\,s^{-1}}$ |
| Wind shear exponent ($\alpha_{\mathrm{PL}}$) | Bimodal normal | Wind speed | $2\,\mathrm{m\,s^{-1}}$ |
| Air density ($\rho_{\mathrm{air}}$) | Bimodal log-normal | – | – |
| Significant wave height ($H_s$) | Gumbel, Weibull | Wind speed | $2\,\mathrm{m\,s^{-1}}$ |
| Wave peak period ($T_p$) | Bimodal Gumbel | Wave height | $0.5\,\mathrm{m}$ |
| Wave direction ($\theta_{\mathrm{wave}}$) | Non-parametric KDE | Wave height and wind direction | $1.0\,\mathrm{m}$ and $30°$ |
| Water density ($\rho_{\mathrm{water}}$) | Trimodal normal | – | – |
| Near-surface current ($v_{\mathrm{NS}}$) | Weibull | – | – |
| Sub-surface current ($v_{\mathrm{SS}}$) | Weibull, Gumbel | – | – |
| Deviation NS direction ($\Delta_{\mathrm{NS}}$) | Bimodal normal | (Wind direction and NS direction) | – |
| SS direction ($\theta_{\mathrm{SS}}$) | Non-parametric KDE | – | – |

$P(T_p) = P(T_p | 1.5\,\mathrm{m} \leq H_s < 2\,\mathrm{m}))$. The bin widths for the dependent parameters are summarised in Table 2 as well. For highly correlated parameters, an alternative to the binning procedure is to model only the deviation between the parameters. Here, the direction of the near-surface current that is highly dependent on the wind direction is an example. Therefore, by modelling the deviation $\Delta_{\mathrm{NS}}$ according to Table 2, it applies:

$$\theta_{\mathrm{NS}} = \Delta_{\mathrm{NS}} + \theta_{\mathrm{wind}} \tag{13}$$

Visual inspections and objective criteria using Kolmogorov-Smirnov tests (KS tests) and chi-squared tests ($\chi^2$ tests) are used to select the best fitting distribution for each environmental condition. Although the KS test is less powerful than other statistical tests, it is still used due to its suitability for small samples (occurring for example for dependent variables and high wind speeds), where $\chi^2$ tests are not applicable. For one parameter, it is attempted to chose only one distribution for all bins and sites in order to keep the data basis easy to use. However, as noted in Table 2, in some cases several distributions are selected to increase the accuracy of the fits.

Directional parameters like $\theta_{wind}$ are treated differently, as classical, parametric distributions can hardly fit several peaks in continuous distributions ($0° = 360°$). Therefore, a non-parametric kernel density estimation (KDE) is used to fit directional parameters.

## 2.3 Resulting distributions

In order to establish a full data basis, statistical distribution and their parameters for all thirteen environmental conditions, the three sites and all bins (if necessary) have to be provided. Furthermore, for non-parametric distributions the underlying





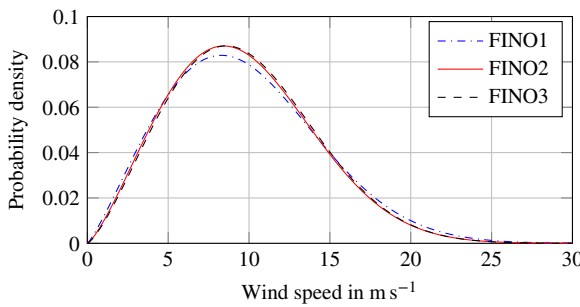

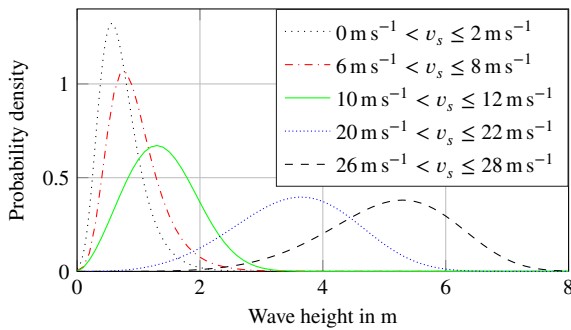

**Figure 4.** Weibull distributions for the wind speeds for all three sites

**Figure 5.** Distribution of the significant wave height for different wind speeds and the FINO1 site. For $v_s \leq 10\,\mathrm{m\,s^{-1}}$, Gumbel distributions are applied. For higher wind speeds, Weibull distributions fit the data more accurately.

data is needed. The main ideas are explained here, however, due to the comprehensiveness of the data, detailed and additional information is provided in an easily applicable form, in the supplementary material. At this point, only two examples are shown in Fig. 4 and 5.

## 2.4 Special findings

In this section, some noteworthy findings of this data basis, mainly resulting from the consideration of scattering, are pointed out. Three examples are presented: the importance of wave peak periods, the high scattering of wind shear exponents, and the behaviour of the turbulence intensity.

Wave loads are of particular importance, if the wave frequency is close to the first natural frequency of the structure. Standard offshore wind turbines have first bending frequencies of about 0.25 to 0.3 Hz (Jonkman and Musial, 2010; Popko et al., 2012) corresponding to eigenperiods of less than 4 s. If state-of-the-art data bases are used (cf. Table 1), there will be no resonance. However, real data suggests that resonance effects are problematic even for higher wind speeds, as wave peak periods of less than 4 s occur (see Fig. 6).

Concerning the wind shear exponent, in the standards and most current data bases (e.g. GL (2012); Fischer et al. (2010)), constant values for all wind speeds are proposed. However, this assumption is a massive simplification. Ernst and Seume (2012) showed that the wind shear exponent significantly depends on the wind speed. Here, it is shown (see Fig. 7) that it does not only vary between wind speeds, but scatters remarkably within each bin as well, and might even be negative.

For the turbulence intensity, this data basis reveals that state-of-the-art approaches are mainly conservative, as too high turbulence intensities are assumed. This is shown in Fig. 8, where the turbulence intensity for all three sites is compared to a standard data basis (Fischer et al., 2010) and to current standards (IEC, 2009). All three sites exhibit similar mean turbulence intensities and 90 % percentile values ($Q_{0.9}$). For the comparison with literature values, the 90 % percentile is of importance, as standards require simulations with this percentile value. However, even for the 90 % percentile, the UPWIND data basis is very





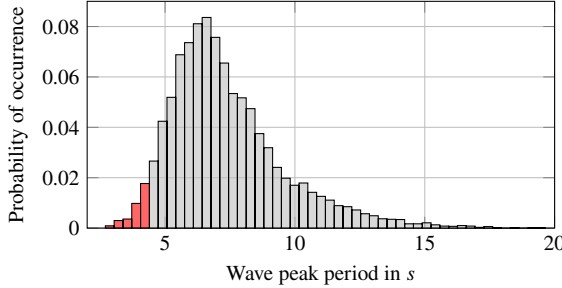

**Figure 6.** Probability distribution of the wave peak period for $v_s = 11\text{-}13\,\mathrm{m\,s^{-1}}$ for the FINO3 site.

**Figure 7.** Distribution of the wind shear exponent for different wind speeds for the FINO2 site.

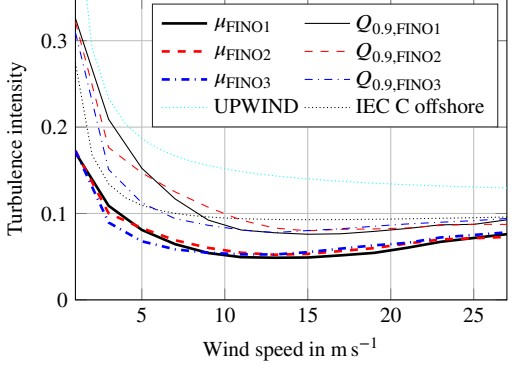

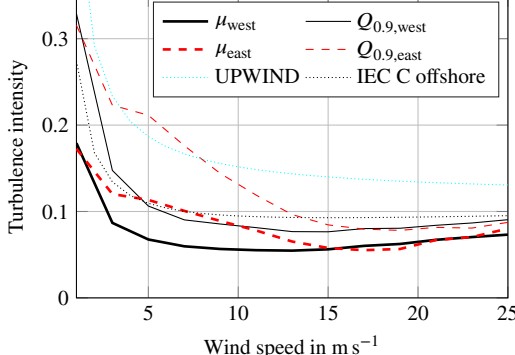

**Figure 8.** Turbulence intensity (mean value and 90 % percentile ($Q_{0.9}$)) for different wind speeds compared to literature.

**Figure 9.** Shadow effects on the turbulence intensity for FINO1 and free stream (western) and wake (eastern) conditions.

conservative. The least conservative case (category C) in IEC (2009) fits the $Q_{0.9}$-values relatively well, but predicts slightly higher turbulence intensities for wind speeds above about $10\,\mathrm{m\,s^{-1}}$. Considering the fact that using the 90 % percentile is a conservative assumption and that the measurements include some wake effects due to wind farms near to all measurement masts, it can be concluded that state-of-the-art assumptions for turbulence intensities are probably unnecessarily conservative. The wake

5    effects are depicted in Fig. 9, where turbulence intensity measurements of FINO1 from 2011 to 2016 are shown. In this period, the wind farm Alpha Ventus was operating on the east side of FINO1. Therefore, west wind leads to free stream conditions and east wind to wake conditions. Obviously, free stream conditions lead to even lower turbulence intensities, whereas wake conditions increase the turbulence especially for smaller wind speeds, as also detected by Hansen et al. (2012).

# 3    Simulation assistance

10    In the previous section, a comprehensive data basis for scattering environmental offshore conditions was developed. However, even with realistic input parameters the accuracy of numerical simulations is significantly influenced by constraints like their




lengths and start-up times. Therefore, in this section, efficient simulation lengths and start-up times for varying wind speeds and different types of loading and substructures are determined. This is achieved by analysing the convergence of relevant quantities (i.e. FLS and ULS loads). Before conducting these studies, the utilised simulation model and the chosen environmental conditions are briefly presented.

## 3.1 Simulation setup

As environmental conditions vary for various turbine sites, a data basis being used for the studies of convergence has to be chosen. The basis developed in this work is appropriate, and the FINO3 site is chosen. Some conditions, like air and water density, are kept fixed, as it was shown that their variation is of minor importance (Hübler et al., 2017). It is tried to keep

the convergence study as simple as possible, and to focus on the most relevant parameters. In addition to the determined distributions of wind speed and direction, wave height, direction and period, turbulence intensity, and wind shear exponent, the following assumptions are made for all simulations:

- The turbulent wind field is computed according to the Kaimal model and using the software TurbSIM (Jonkman, 2009).

- Irregular waves are calculated according to the Pierson-Moskowitz spectrum.

- Soil conditions of the OC3 model (Jonkman and Musial, 2010) are applied.

- The current, second-order and breaking waves, wave spreading effects, marine growth, local vibration effects of braces, joint stiffnesses, and degradation effects are neglected.

The time domain simulations of the convergence study are conducted using the aero-servo-hydro-elastic simulation framework FASTv8 (Jonkman, 2013). A soil model (Häfele et al., 2016) applying linearised soil-structure interaction matrices enhances

this code. The NREL 5 MW reference wind turbine (Jonkman et al., 2009) with two different substructures is investigated: Firstly, the OC3 monopile (Jonkman and Musial, 2010) and secondly, the OC4 jacket (Vorpahl et al., 2013).

Since the convergence of fatigue and ultimate loads is investigated in the next step, the calculation concept of these two loads is briefly explained. The outcomes of the FAST simulations are, inter alia, time series of forces, moments, and stresses for each element of the substructure. For the fatigue analysis, hot spot stresses have to be calculated using these time series. For the

monopile, this is done according to Eurocode 3, part 1-9 (2010), where a detail of 71 MPa for butt welds and an additional reduction due to a potential eccentricity is recommended. For the jacket, DNV-RP-C203 (2010) is applied. Hot spots are analysed and stress concentration factors (SCF) depending on the joint geometry are used. The fatigue limit is 52.6 MPa at $10^7$ cycles. This corresponds to the DNV-GL S-N curve 90 (for cathodic protection) as used in the original design (Vemula et al., 2010). For all stresses, a Rainflow counting evaluates the stress cycles. As recommended by the current standards, the conservative

damage accumulation according to the Palmgren-Miner rule is assumed using a slope of the S-N curve of three before and five after the fatigue limit for both substructures. For the ULS analysis, maximum stresses are decisive and extracted from the time series. For the monopile, Eurocode 3, part 1-6 (2010), is used to analyse the plastic limit state, cyclic plasticity limit state, and





buckling limit state (LS1-3). For the jacket, NORSOK N-004 is applied for tubular members and joints which takes combined axial, shear, bending and hydrostatic loadings into account. In both cases, the yield stress is 355 MPa. Additionally, ultimate limit state proofs for the foundation piles are performed including axial and lateral soil proofs according to GEO2 (DIN 1054, 2010) and a plastic limit state proof (LS1) for the steel pile below mudline. Especially for the monopile, the last proof might be

decisive as the bending moment frequently reaches its maximum below mudline. For all ULS proofs, utilisation factors, being the percentage of the maximum loads, are the outcomes.

## 3.2   Simulation length

The simulation length significantly influences the overall computing time of the load assessment. However, there is no conclu-

sive consensus concerning the length needed. Current standards recommend for example 10-minute or one-hour calculations. The offshore oil and gas industry prefers simulation lengths of six hours to cover all low-frequency hydrodynamic effects.

The use of 10-minute simulations can potentially reduce the computing time by a factor of about 36 compared to six-hour simulations. Hence, a study of convergence for bottom fixed offshore wind turbines is conducted here. For floating wind turbines, it is referred to Stewart (2016), who showed that for floating structures all physical effects can be covered with 10-minute

simulations.

The presented outcomes of this study focus on the monopile substructure, but results for the jacket (not shown) are generally comparable. For several wind speed bins, 500 simulations with a total length of ten hours are conducted. As the start-up behaviour is analysed subsequently, a clearly sufficient start-up time of four hours is chosen. Discarding the start-up time, the simulation length of 10 h reduces to a maximum available length of 6 h for the convergence study. In a first step, the conver-

gence of FLS loads is analysed. Afterwards, the ULS case is investigated.

Figure 10 displays the normalised mean fatigue damages for different wind speeds and simulation lengths between ten minutes and six hours. The values are normalised with the six-hour values, and error bars show the $\pm\sigma$ confidence intervals (68 %) that are estimated using a bootstrap procedure with 10 000 resamplings.

It is apparent that due to scattering environmental conditions and the limited number of simulations the uncertainty is relatively

high. A detailed investigation of the fatigue load uncertainty, when scattering environmental conditions are applied, is valuable, but out of the scope of this work (cf. Sec. 4). Nevertheless, from Fig. 10 it is apparent that there are no pronounced trends for changing simulation lengths. A slight increase of fatigue loads for higher simulation lengths might be suspected given the fact that such behaviour was observed for floating substructures by Stewart (2016). In order to focus on the simulation length effects, the variation of environmental conditions is neglected in a second step. This reduces the uncertainty making it possible

to clearly identify a slight increase of FLS loads of about 5 % for higher simulations lengths (see Fig. 11, not merge case). However, as shown by Stewart (2016) for floating substructures, the increasing fatigue loads are not due to any physical effect (all important low-frequency effects of waves are already covered by 10-minute simulations), but can be explained by the effect of unclosed cycles in the Rainflow counting. Cycles that are not completed at the end of the simulation are approximated by counting them as half cycles. The longer the simulation, the less influential is this approximation. A quite straightforward ap-





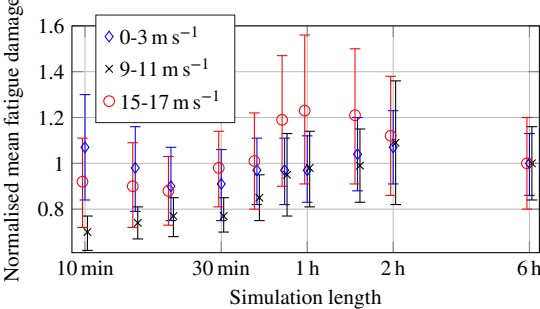
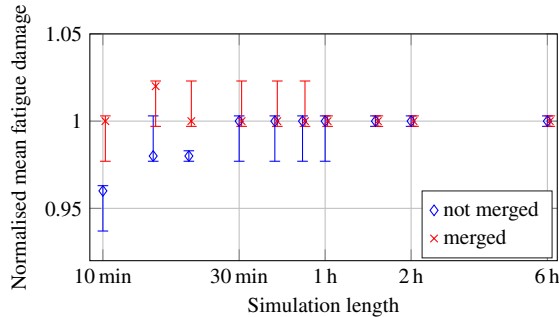

**Figure 10.** Normalised mean fatigue damage (500 simulations) for increasing simulation lengths and different wind speeds.

**Figure 11.** Normalised mean fatigue damage (500 simulations) for increasing simulation lengths and $v_s = 9\text{-}11\,\mathrm{m\,s^{-1}}$. Environmental conditions are kept constant to demonstrate the effect of merging time series more clearly.

proach to reduce the problem of half cycles is to merge several shorter simulations to a longer one. It is possible to merge either different 10-minute simulations, or each time series is duplicated and merged with itself. If scattering environmental conditions are assumed, fairly different load levels in some simulations occur. In these cases, load levels of the simulations might not fit, and additional cycles can be introduced by using the first approach, leading to unreasonably increased fatigue damages. The

second approach guarantees fitting load levels by utilising one and the same simulation. On the downside, the computing time of the post-processing is increased. The effect of merging several 10-minute simulations with itself (only shown for constant environmental conditions) is demonstrated in Fig. 11. Hence, it is possible to compensate the simulation error of short simulations by merging time series in the post-processing.

For the ULS loads, the convergence is shown in Fig. 12. Obviously, ULS loads are higher for longer simulations. Again, this

increase is not due to any physical phenomenon, but a result of different overall computing times. Clearly, 500 10-minute simulations should not be compared to 500 six-hour simulations, but to about 14 six-hour simulations (Haid et al., 2013). This comparison is displayed in Fig. 13 and makes clear that ULS loads do not depend on the simulation length but only on the overall computing time. A second fact being visible in Fig. 13 are the higher uncertainties for longer simulation lengths. Since 10-minute simulations lead to a higher number of cases than six-hour simulations for the same total length (i.e. 500 and 14),

shorter simulations better cover rare cases, and therefore, scattering environmental conditions leading to less uncertainty.

After all, the investigations of this section suggest that simulations of ten minutes length are sufficient independent of the type of load or substructure, or wind speed. For ULS loads, the same overall time has to be compared in order to achieve reliable results. By keeping the simulation length short, more simulations can be conducted in the same overall computing time leading to a better convergence of ULS loads. For FLS loads, simulation errors due to the simulation length can be reduced by merging

the time series.





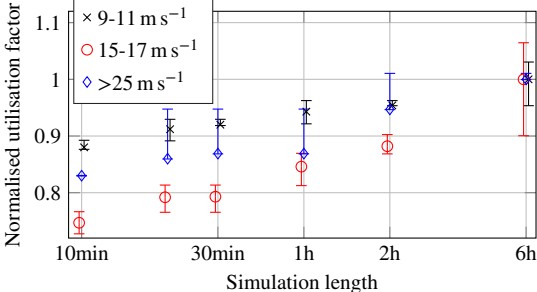
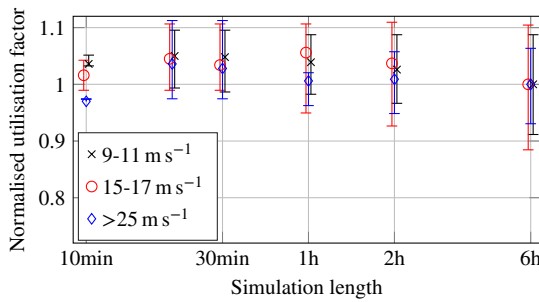

**Figure 12.** Normalised mean ULS utilisation factor (500 simulations) for increasing simulation lengths and different wind speeds.

**Figure 13.** Normalised mean ULS utilisation factor for increasing simulation lengths (constant overall length of $500 \times 10$ min leading to 500 to 14 simulations) and different wind speeds.

### 3.3 Start-up time

For the analysis of the simulation length, start-up times of four hours were used to guarantee a steady state operation of the turbine. However, the usage of four hours of start-up and only ten minutes of simulation is computationally very expensive. Therefore, the convergence of FLS and ULS loads with respect to the start-up time is analysed. As initial conditions, like an

initial rotor speed, influence the transient start-up behaviour (Haid et al., 2013), initial rotor speeds and blade pitches depending on the wind speed are set here. These initial conditions are quasi-static states determined using prior simulations.

As the start-up behaviour is affected by the type of substructure and the load condition, the start-up time is analysed in each wind speed bin for FLS and ULS loads and for both types of substructures separately. Commonly, time series are investigated to estimate start-up times (Zwick and Muskulus, 2015). Although this is a straightforward approach, here, it is considered to be

not expedient. For a fatigue assessment, the convergence of the fatigue damage has to be analysed, and for the ULS analysis, maximum loads or utilisation factors have be considered.

For each wind speed bin, 10 000 simulations for the monopile and 500 for the jacket were conducted. The high and unequal number of simulations is needed to exclude effects of the number of simulations, mentioned in the previous section and addressed in Sec. 4, as well as possible. For the monopile, each simulation at operating conditions is 900 s long (600 s simulation

length plus 300 s start-up to reach the steady state) and 1800 at idling conditions. When the turbine is idling, the aerodynamic damping is lower, leading to higher start-up times. For the jacket, all simulations are 720 s long. Using this simulated data basis, it is possible to analyse the effect of different start-up times up to 300 s (1200 s for idling; 120 s for the jacket) on the fatigue damage and utilisation factors in order to determine optimal start-up times.

Figure 14 displays the convergence of the fatigue damage of the monopile substructure at operating conditions. For idling

conditions (not shown), the start-up takes longer, as the aerodynamic damping is lower. For the same reason, the transients are shorter for higher wind speeds. For the jacket substructure displayed in Fig. 15, the transients decay much faster in all wind speed bins. As jackets are less influenced by wave loads, being not always aligned with the wind, the aerodynamically marginally damped side-to-side modes are less excited, leading to a shorter transient behaviour. This interpretation is supported





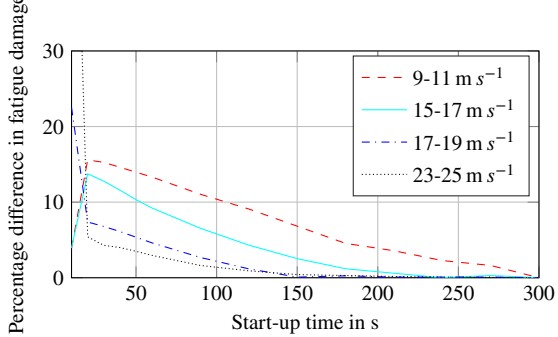
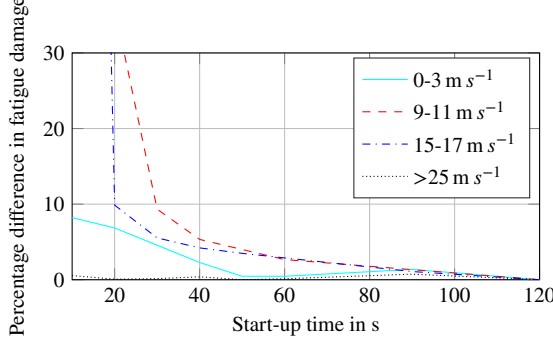

**Figure 14.** Start-up behaviour of the operating wind turbine with a monopile substructure for different wind speeds. Percentage difference in the fatigue damage compared to the "converged" value (300 s).

**Figure 15.** Start-up behaviour of the wind turbine with a jacket substructure for different wind speeds. Percentage difference in the fatigue damage compared to the "converged" value (120 s).

**Table 3.** Recommended start-up times for simulations with OC4 jacket and OC3 monopile substructures for different wind speeds to achieve errors below 5 %.

| $v_s$ in m s$^{-1}$ | Case | < 3 | 3-5 | 5-7 | 7-9 | 9-11 | 11-13 | 13-15 | 15-17 | 17-19 | 19-21 | 21-23 | 23-25 | > 25 |
|---|---|---|---|---|---|---|---|---|---|---|---|---|---|---|
| Monopile | FLS | 720 s | 240 s | 240 s | 240 s | 240 s | 240 s | 240 s | 150 s | 120 s | 60 s | 60 s | 60 s | 360 s |
| Jacket | | 40 s | 30 s | 50 s | 40 s | 50 s | 50 s | 50 s | 50 s | 50 s | 60 s | 50 s | 50 s | 10 s |
| Monopile | ULS | <10 s | <10 s | <10 s | <10 s | 10 s | 10 s | 10 s | 10 s | 10 s | 10 s | 10 s | 10 s | <10 s |
| Jacket | | <10 s | 20 s | 20 s | 20 s | 20 s | 20 s | 20 s | 20 s | 20 s | 20 s | 20 s | 20 s | <10 s |

by the fact that for the jacket, idling conditions, where the hydrodynamic behaviour dominates, have short start-up times.

The convergence of ULS utilisation factors for both substructures is shown in Fig. 16 and 17. It becomes apparent that short start-up times are needed independent of the type of substructure and wind speed. The cycles with high amplitudes occurring during the start-up are damped out within a few seconds, and hence, are not influencing the ULS behaviour. More problematic are less damped cycles with smaller amplitudes leading to the previously presented, higher start-up times for FLS loads.

The recommended start-up times for both substructures, being always a compromise between computing time and accuracy (here, errors below 5 %), are summarised in Table 3. It has to be mentioned that a general validity was not checked, and start-up times might be slightly varying for substantially different substructures. For example, jackets for 10 MW turbines might behave differently due to larger diameters of legs and braces increasing wave effects. However, the given values represent a well-founded guidance for first simulation set-ups.



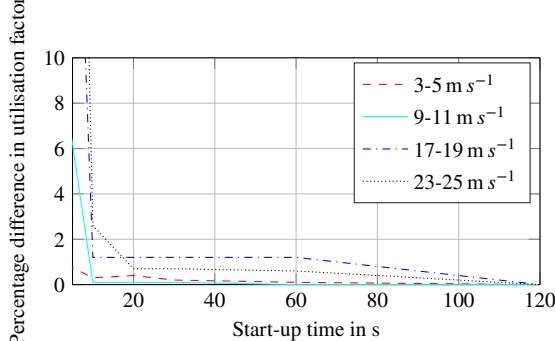

**Figure 16.** Start-up behaviour of the wind turbine with a monopile substructure for different wind speeds. Percentage difference in the utilisation factor (ULS) compared to the "converged" value (300 s).

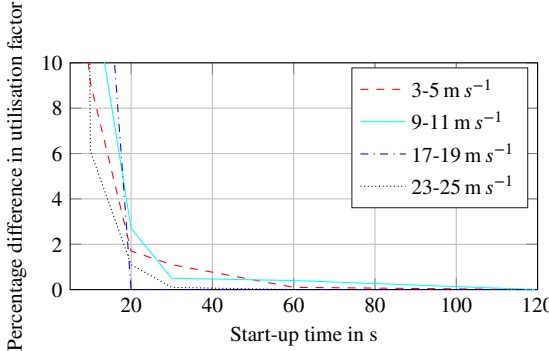

**Figure 17.** Start-up behaviour of the wind turbine with a jacket substructure for different wind speeds. Percentage difference in the utilisation factor (ULS) compared to the "converged" value (120 s).

## 4   Benefits and limitations

The benefit of the current work is twofold. Firstly, a comprehensive data basis for scattering environmental conditions was set up, which is freely available and easy to use. Secondly, two simulation constraints (simulation length and start-up time) were analysed, and well-founded recommendations are given.

The main advantages over existing data bases are the following: The data basis covers several different sites being situated in different oceans. It has to be admitted that the sites are fairly similar, as they are all in shallow water conditions. Additionally, the data basis contains statistical distribution for much more environmental conditions than existing ones. As was shown for example by Hübler et al. (2017) that not only main conditions like the wind speed are influencing the dynamic behaviour of offshore wind turbines, knowledge of additional parameters is beneficial. Current data bases consist frequently of raw data that

needs to be post-processed, which is a time-consuming process. Here, on the one hand, easily applicable statistical distributions are given. One the other hand, the complexity of dependent environmental conditions is still covered by utilising conditional distributions and multimodal and non-parametric approaches. In contrast to many existing data bases, the raw data is of good quality. For example, wind speeds are measured at heights comparable to hub heights of current turbines, and there is no need for extrapolations, as it is the case for buoy measurements. Still, more data would be valuable in order to achieve more reliable

distributions in high wind speed bins that rarely occur. After all, the developed data basis is capable to improve offshore wind turbine modelling by providing more realistic inputs for simulations in academia where real site data is scarce. One example of improved offshore wind turbine modelling is given in Sec. 3.2 and 3.3. The inclusion of probabilistic inputs leads to a significant and realistic increase of fatigue damage scattering requiring high numbers of simulations. Hence, deterministic inputs underestimating this scattering can lead to biased fatigue values. Detailed analyses of the effect of scattering environmental

conditions on fatigue damage, and therefore, of the needed number of simulations are part of upcoming work of the authors. Concerning the second benefit, the simulation constraints, it has to be kept in mind that not only realistic modelling, but also





small simulation errors are important in order to model accurately. In this context, the chosen simulation length and start-up time matter. So far, these values are frequently chosen without profound knowledge. Some approaches to gain a deeper insight into these constraints (Stewart, 2016; Zwick and Muskulus, 2015) concentrate on simulation lengths or specific types of substructures and are not taking realistically scattering environmental conditions into account. In this work, the scattering of the

conditions is addressed and different bottom fixed substructures are analysed. This enables recommendations for simulation lengths and start-up times depending on the wind speed, the type of substructure and the considered load case (ULS or FLS). However, the general validity of the current results has to be slightly restricted, as only one design of each type of substructure was investigated. Therefore, start-up times might be slightly different for significantly different designs. Furthermore, for the start-up time, the values might also differ between different simulation codes and are only tested for the FASTv8 code.

Nevertheless, even in these cases, the given recommendations can be regarded as a well-founded starting point for further investigations.

## 5   Conclusions

This work aims to help future simulation work to be more realistic and accurate. In order to achieve this objective, a freely available and comprehensive data basis for scattering environmental conditions was set up. This data basis consists of condi-

tional statistical distribution for many parameters and can be applied without further post-processing. All needed information is given in the supplementary material. Additionally, scientifically sound recommendations are given for the choice of simulation lengths and start-up times. Simulation lengths of 10 minutes are generally sufficient, and can even help to reduce uncertainties. However, in case of FLS loads, times series should be merged, and for ULS situations, the overall computing time has to be kept constant. Recommendations for start-up times are summarised in Table 3. These values can help to improve the accuracy

of simulations, and to reduce computing times. It should be noted that the given start-up times partly differ significantly from values chosen in literature that are mainly based on educated guesses.

An enlargement of the current data basis to include additional offshore sites, other types or designs of substructures or investigations for other simulation codes would be definitely valuable.

*Data availability.*   The raw data is taken from the FINO platforms - operated on behalf of the Federal Ministry for the Environment, Nature

Conservation, Building and Nuclear Safety (BMUB) - and is freely available for research purposes (www.fino-offshore.de/en/). The derived data basis, consisting of statistical distribution for thirteen partly dependent environmental conditions and three offshore sites, is freely available. All needed data is given in the supplementary material to this work.

*Competing interests.*   The authors declare that they have no conflict of interest.



*Acknowledgements.* We gratefully acknowledge the financial support of the Lower Saxony Ministry of Science and Culture (research project VENTUS EFFICIENS, FKZ ZN3024) and the European Commission (research project IRPWIND, funded from the European Union's Seventh Framework Programme for research, technological development, and demonstration under grant agreement number 609795) that enabled this work. This work was supported by the compute cluster which is funded by Leibniz Universität Hannover, the Lower Saxony Ministry of Science and Culture (MWK), and the German Research Foundation (DFG).



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
