# Peer review of "Development of a comprehensive data basis of scattering environmental conditions and simulation constraints for offshore wind turbines"

_Wind Energy Science, 2017_

## Referee Comment (RC1) · Anonymous Referee #1 · 24 Jul 2017

Please see attachment for comments.

Please also note the supplement to this comment:
https://www.wind-energ-sci-discuss.net/wes-2017-24/wes-2017-24-RC1-supplement.pdf

---

## Referee Comment (RC2) · Anonymous Referee #2 · 10 Aug 2017

The manuscript analyzes the measured wind/wave records at three offshore sites Fino 1, 2, 3 to characterize the variation in environmental parameters and then proceeds to examine the implications on the fatigue damage equivalent loads on a 5 MW offshore wind turbine. Analysis of the impact of load simulation time and initial start-up transience on the fatigue computations are examined. While the subject matter of the manuscript is important, there are a number of shortcomings in the content and explanations that need to be corrected as in:

1) A number of environmental variables including air and water density, currents etc.

are captured in section 2 for the 3 offshore sites. However section 3 on the analysis of load simulations essentially investigates simulation length time and initial transience. As it stands, section 2 and section 3 are not well connected and there needs to be a clear explanation made as to how the varying environmental conditions modeled in section 2 are used in the load simulations. Without this connectivity, the paper cannot be published.

2) Similarly to point 1 above, about 2 pages of the manuscript are devoted to analysis of ocean currents, but ocean currents are not used in fatigue load simulations as per the IEC 61400-3 and may have only limited influence on extreme loads. So this analysis on pg. 6-7 can be deleted, unless shown in section 3 to be relevant.

3) Figure 7 on the probability of the wind shear exponent is not clear and it is not evident why the probability of a higher wind shear exponent is greater for higher mean wind speed bins. It would be more appropriate, if the shear exponent probability is plotted for different atmospheric stability classes.

4) In Section 3.1, it is not at all clear how the fatigue damage in welded joints of the monopile and jacket are computed. How are the stress concentration factors at the welded joint computed? How is the circumferential variation of the wind direction over a year modeled especially for the simulation on jackets? What type of jacket joints are considered - K joint, Y joint etc? Without these details, the analysis of fatigue on sub structures is inadequate and incomplete.

5) What load case is analyzed in section 3.2 to compute fatigue damage? Is it only DLC 1.2? What about DLC 6.4, DLC 7.2, 4.1 etc?

6) In Figure 10 and 11, is it the fatigue damage that is plotted or the damage equivalent load?

7) It is not clear how the half-cycles are merged in Fig. 11 and why the variation in fatigue damage suddenly disappears above 1-hour of simulation.

8)Explain what load cases are simulated for ULS loads in Fig 12 and 13 and What is the annual return probability of the ULS loads computed?

9) The start-up time for load simulations depends on the time constants of the aeroelastic models, the frequencies of the turbine and the numerical solver used, besides the damping that is referred to in the paper. So table 3 is highly aeroelastic code and turbine model dependent and cannot be used as a general recommendation.

Overall the paper is presenting results without appropriate explanation of the load cases used, the limitations of the analysis, the justification of the methods used and the underlying assumptions. It needs to be re-written to provide clear and relevant justification of the results and methods.

---

## Author Comment (AC1) · 10 Aug 2017

Please see attachment for comments.

Please also note the supplement to this comment:
https://www.wind-energ-sci-discuss.net/wes-2017-24/wes-2017-24-AC1-supplement.pdf

---

## Author Comment (AC2) · 17 Aug 2017

**Response to reviewer's comments:**

We want to thank the reviewer for the detailed and critical review. The valuable comments have improved the quality of the paper by making the objective, the probabilistic simulation approach, and the connectivity of the two parts much clearer. Furthermore, the comments helped us to re-write the paper in order to provide more precise information on the methods used and the underlying assumptions.

In the following, you can find our answers to the comments. A revised version of the manuscript is prepared and the corresponding paragraphs are marked in the manuscript. Pages and lines used in the answers refer to the new version.

*General response concerning the overall simulation approach and the main objectives:*

First of all, we want to give a general response concerning the overall (probabilistic) simulation approach in section 3 and the main objective of this work, as both is obviously not clear enough. These explanations are now included in the paper section 3.1 to make it clearer to the reader.

The main objective of this work is to provide a data basis for scattering environmental conditions and to give practical recommendations. Both is intended to make future simulations more realistic and reliable. We do not want to present a precise methodology to calculate ULS and FLS loads for turbine design or optimisation.

Many simulation approaches in academia focus on power production load cases (e.g. [1-3]), as other load cases are, firstly, frequently less relevant (e.g. for the considered jacket [4]), secondly, are very controller and design dependent, and thirdly, need special treatment (e.g. simulation lengths and initial transient times can be completely different). Therefore, this work intends to give guidance on simulation lengths and initial transient times for future work in academia focussing on power production, if scattering environmental conditions are applied.

This main objective leads to three implications:

a) There is no need of "perfect" FLS/ULS calculation, as the objective is not the computation of the lifetime of the turbine or a new design (we do not perform turbine design according to the standards), but the investigation of simulation constraints.

b) Consequently, there is no need to simulate all load cases. Start up, shut down and other "special" load cases need special treatment anyway (simulation constraints will be different compared to power production load cases), and therefore, are not investigated here.

c) The approach of using scattering environmental conditions leads to "real-life" simulation and not a "load case based" simulation. This means that the simulation shall represent the real lifetime of the turbine (without fault, start-up, etc.). Hence, the simulations (e.g. 10000 simulations) cover a period of power production and idling, leading to 2.3 months of turbine lifetime (for 10000 simulations). As environmental conditions scatter, effects like high turbulences, extreme wind shear, high waves, small wave periods, and others are covered and do not have to be considered separately. Load cases are not simulated explicitly, but are cover implicitly by conducting probabilistic simulations.

That is why for FLS, the two approaches do not significantly differ. The "real-life" approach covers DLC 1.2 and 6.4. For ULS, the "real-life" approach covers all power production cases (DLC 1.1-1.6) and DLC 6.1 by applying scattering environmental conditions. As the "real-life" approach cannot simulate 20 years of turbine lifetime (or even a return period of 50 years), a load extrapolation, as required for DLC 1.1, is needed in order to calculate a "perfect" ULS design. However, this extrapolation is not needed here, as it does not influence the investigated simulation constraints.

Whether this probabilistic "real-life" approach represents the turbine life correctly is not investigated here, as this is not of primary significance (see a)). In any case, simulation constraints for power production load cases can be analysed with this procedure, as all these load cases are covered.

1.) *A number of environmental variables including air and water density, currents etc. are captured in section 2 for the 3 offshore sites. However section 3 on the analysis of load simulations essentially investigates simulation length time and initial transience. As it stands, section 2 and section 3 are not well connected and there needs to be a clear explanation made as to how the varying environmental conditions modeled in section 2 are used in the load simulations. Without this connectivity, the paper cannot be published.*

You are right, that, as it stands, it was not clear enough that scattering environmental conditions, being derived in section 2, are applied for all simulations in section 3. This should now be much clearer. Firstly, in section 3.2, the application of statistically scattering values in all simulations is now mentioned. Secondly, section 3 was complemented by a detailed explanation of the overall simulation approach according to the "general response" above (section 3.1). This makes the use of scattering environmental conditions in the "real-life" approach clear, and therefore, connects the two sections.

2.) *Similarly to point 1 above, about 2 pages of the manuscript are devoted to analysis of ocean currents, but ocean currents are not used in fatigue load simulations as per the IEC 61400-3 and may have only limited influence on extreme loads. So this analysis on pg. 6-7 can be deleted, unless shown in section 3 to be relevant.*

It is correct that the influence of ocean currents is limited and not used for the FLS simulations according to IEC 61400-3. However, due to several reasons, the authors regard the inclusion of statistical distribution for ocean currents as beneficial. Firstly, according to IEC 61400-3, ocean currents have to be included in the ULS calculation. Therefore, for these load cases a statistical, more realistic representation of ocean currents might be valuable. Secondly, although the influence of currents on ULS loads is limited, it is not neglectable [5]. Thirdly, the current data basis is intendent to be slightly more general and hopefully relevant for some other applications as well. Hence, even if ocean currents would not influence ULS loads (which is not the case), the inclusion of ocean currents in the data basis might be useful for application concentrating on hydrodynamic effects.

Therefore, in section 3, it does not have to be shown that ocean currents are relevant for FLS or ULS loads. Sensitivity analyses for all parameters are out of the scope of this paper, but could use the provided statistical distributions [5].

3.) *Figure 7 on the probability of the wind shear exponent is not clear and it is not evident why the probability of a higher wind shear exponent is greater for higher mean wind speed bins. It would be more appropriate, if the shear exponent probability is plotted for different atmospheric stability classes.*

It is right that a correlation between the shear exponent and the atmospheric stability is probably more common. However, there is also a correlation between atmospheric stability and wind speed with higher stabilities for higher wind speeds [6]. Therefore, it is also possible to correlate wind speed and wind shear.

Additionally, the distributions given in Fig. 7 are based on real data. If you consider the following figures, you can firstly see the real data histograms for three different wind speeds. Secondly, there is a correlation plot for the wind speed and the wind shear exponent. Therefore, it is evident that, for the considered sites, higher wind shear exponents occur at higher wind speeds. This feature has also been detected in [6, 7] for example.

A phenomenological investigation of the reasons for this feature is out of the scope of this work, as this work is data based.

[Figure]

Real data histograms of wind shear exponents depending on the wind speed (FINO2 data)

[Figure]

Correlation plot: Wind speed versus for wind shear exponents with increasing mean wind shear (For reasons of clarity, only one week of FINO2 data is used. The use of all data features the same correlation, but due to more than 100 000 data points the plot gets unclear)

*4.) In Section 3.1, it is not at all clear how the fatigue damage in welded joints of the monopile and jacket are computed. How are the stress concentration factors at the welded joint computed? How is the circumferential variation of the wind direction over a year modeled especially for the simulation on jackets? What type of jacket joints are considered - K joint, Y joint etc? Without these details, the analysis of fatigue on sub structures is inadequate and incomplete.*

The explanations in section 3.2 (formerly 3.1) are definitively too brief to be clear. Therefore, the paragraph on fatigue calculation using current standards is improved by giving more details.

First, the procedure is clarified: Due to the statistical variation of the environmental conditions that is applied, the conducted simulations represent a typical part of the turbines lifetime (without failure and start-up cases etc.). For example, 10000 simulations represent about 2.3 months of realistic turbine lifetime. Directional variations of the wind are already included by just applying the scattering environmental conditions. Hence, summing up the damages of all ten-minute simulations gives a realistic estimate of the damage occurring during this time (e.g. 2.3 months). Surely, the damages in each joint and even each connection of the joint are different, and the highest damages do not occur in the same joint connection for each simulation (for example due to the variation in the wind direction). Therefore, for the connections of all joints (K-joints, Y-joints, butt-welds, …) the summation over all simulations is done separately, leading to overall damages (2.3 month damages) for each joint connection. Now, the highest accumulated damage value of all joint connections is used to be the jacket damage as it is the most critical one.

Second, the damage calculation for each simulation is clarified: For the jacket, DNV-RP-C203 is applied. For each connection to a joint, eight spots around the circumference of the intersection are calculated according to section 3.3.2 (of DNV-RP-C203). The needed SCFs are computed according to Appendix B for different joint geometries. For the monopile, Eurocode 3, part 1-9 is used. Guidance on stress producing effects of details and welds is given in Table 8.1 to 8.10. For the monopile, a detail of 71 MPa for transverse butt welds (longitudinal welds have higher details, and therefore, are less critical) is used. Furthermore, an additional reduction due to the size effect (t>25mm, c.f. Table 8.3) is applied.

*5.) What load case is analyzed in section 3.2 to compute fatigue damage? Is it only DLC 1.2? What about DLC 6.4, DLC 7.2, 4.1 etc?*

As stated in the general response, for fatigue, the probabilistic approach covers implicitly DLC 1.2 and 6.4. Other, "special" load cases are not analysed due to the three already mentioned reasons: Limited relevance (e.g. for the jacket [4]), controller and design dependence, special treatment due to different simulations constraints (e.g. initial transients are relevant for physical turbine start-ups (DLC3)), and no "perfect" fatigue calculation needed.

6.) *In Figure 10 and 11, is it the fatigue damage that is plotted or the damage equivalent load?*

Figure 10 and 11 show the mean fatigue damage that is normalised with the six-hour values as stated on p.14. The procedure how these values are calculated should now be explained much clearer (additional section on the overall simulation setup (section 3.1), and paragraphs on the fatigue model (p.12-13), and on the statistical calculation procedure (p.13-14)) .

*7.) It is not clear how the half-cycles are merged in Fig. 11 and why the variation in fatigue damage suddenly disappears above 1-hour of simulation.*

The procedure of merging several times series to eliminate the effects of half cycles is explained on p.14-15 and is now further elaborated. In Fig. 11 "not merged" case, it can be seen that, for simulation lengths below about two hours, the normalised mean fatigue damage is lower than for the converged case (six hours simulation length). This is due to unclosed half cycles in each time series. For long simulations, the effect of half cycles can be neglected, as the number of unclosed cycles is small compared to the number

of closed cycles. Therefore, it is possible to eliminate this effect in short simulations (e.g. ten-minute simulations) by merging each short simulations several times with itself. This means: Each ten-minute time series is duplicated and then appended several times to itself until a six-hour time series is formed consisting of 36 identical ten-minute time series. For this generic and repeating six-hour time series, the ratio of unclosed cycles to closed cycles is small enough in order to eliminate the effect of half cycles.

The reason for disappearing variations in fatigue damages for longer simulations is twofold. Firstly, in Fig. 11, there was a numerical problem. Due to too few significant numbers used, small variations were rounded to zero. This is now removed and Fig. 11 is revised. However, secondly, the uncertainty actually reduces for longer simulations. This is due to the fact that the overall length (simulation length multiplied by number of simulations) increases. This effect of not keeping the overall length constant is - in another context - discussed in section 3.3 for ULS loads (Fig. 12 and 13).

8.) *Explain what load cases are simulated for ULS loads in Fig 12 and 13 and what is the annual return probability of the ULS loads computed?*

Similar to remark 5), the simulated load cases are explained in the general response and are now included in section 3.1. Load cases according to the standards are not simulated separately, but a "real-life" approach is applied. "Special" load cases (start-up, etc.) are not included. For power production and idling, the "real-life" approach covers DLC 1.1, DLCs 1.3 to 1.6, and DLC 6.1 as environmental conditions are varied. Consequently, the conducted simulations do not represent classical extreme event simulations with specific return periods, but act as a realistic lifetime period (e.g. 2.3 months for 10000 simulations) that can be extrapolated to 20 or 50 years, by applying extrapolation methods like those for DLC1.1. This extrapolation is not done here, as the focus of this work is not on ULS values, but on simulations constraints for ULS simulations that are not influenced by the extrapolation.

9.) *The start-up time for load simulations depends on the time constants of the aeroelastic models, the frequencies of the turbine and the numerical solver used, besides the damping that is referred to in the paper. So table 3 is highly aeroelastic code and turbine model dependent and cannot be used as a general recommendation.*

So far, it was not explicitly stated that Table 3 cannot be used as a general recommendation. This was now revised, and the additional limitations you mentioned are included. However, these values are still relevant due to two reasons: Firstly, for similar applications (FASTv8, NREL 5MW turbine, etc.) that are not rare in academia (e.g. [1, 2, 8, 9]) the values might be used. And secondly, even more important, these results can sensitise other researches to the problem of initial transients especially in case of fatigue. For fatigue, times of initial transients might be higher than expected or applied in literature, and a pure glance on time series is not sufficient, as less damped cycles with small amplitudes cannot directly be seen/identified.

*Overall the paper is presenting results without appropriate explanation of the load cases used, the limitations of the analysis, the justification of the methods used and the underlying assumptions. It needs to be re-written to provide clear and relevant justification of the results and methods.*

The paper was rewritten in order to provide a much clearer presentation of the methods used and of their limitation. Especially, section 3 was enhanced by an extensive explanation of the overall simulation approach (section 3.1). This extension of the paper clarifies the load cases covered without directly simulating load cases according to the standards, but applying a probabilistic approach. Furthermore, it makes clear the limitations concerning the non-covered load cases and the FLS and ULS calculations that are not "perfect" and not suitable for a design procedure. However, it justifies why, for the present objective of giving recommendations for simulation constraints, this approach is sufficient.

For other aspects (e.g. wind shear, ocean currents), justifications for the utilized methods are given in this response to the comments.

The limitation concerning the general validity of times of initial transients are stated more precisely, but the remaining benefit is highlighted as well.

[1] Morató, A., Sriramula, S., Krishnan, N., & Nichols, J. (2017). Ultimate loads and response analysis of a monopile supported offshore wind turbine using fully coupled simulation. *Renewable Energy*, *101*, 126-143.

[2] Zwick, D., & Muskulus, M. (2015). The simulation error caused by input loading variability in offshore wind turbine structural analysis. *Wind energy*, *18*(8), 1421-1432.

[3] Haid, L., Stewart, G., Jonkman, J., Robertson, A., Lackner, M., & Matha, D. (2013, June). Simulation-length requirements in the loads analysis of offshore floating wind turbines. In *ASME 2013 32nd International Conference on Ocean, Offshore and Arctic Engineering* (pp. V008T09A091-V008T09A091). American Society of Mechanical Engineers.

[4] N.K. Vermula, Deliverable D4.2.5-WP4.2: Offshore Foundations and Support Structures, UpWind Project, 2010.

[5] Hübler, C., Müller, F., Gebhardt, C. G., & Rolfes, R. (2017). Global Sensitivity Analysis of Offshore Wind Turbine Substructures. In *Proceedings of the 15$^{th}$ International Probabilistic Workshop*, accepted for publication.

 [6] Holtslag, M. C., Bierbooms, W. A. A. M., & Van Bussel, G. J. W. (2014). Estimating atmospheric stability from observations and correcting wind shear models accordingly. In *Journal of Physics: Conference Series* (Vol. 555, No. 1, p. 012052). IOP Publishing.

[7] Ernst, B., & Seume, J. R. (2012). Investigation of site-specific wind field parameters and their effect on loads of offshore wind turbines. *Energies*, *5*(10), 3835-3855.

[8] Ziegler, L., & Muskulus, M. (2016, September). Fatigue reassessment for lifetime extension of offshore wind monopile substructures. In *Journal of Physics: Conference Series* (Vol. 753, No. 9, p. 092010). IOP Publishing.

[9] Amirinia, G., & Jung, S. (2017). Buffeting response analysis of offshore wind turbines subjected to hurricanes. *Ocean Engineering*, *141*, 1-11.

[revised manuscript text omitted]

(2) Give well-founded guidance on simulation length requirements and the time needed to exclude initial transients, when these realistic conditions are applied, to improve accuracy of numerical simulations.

In order to address these topics, firstly, a data basis for all significant environmental conditions is derived from real data of the
15  FINO research platforms. In this work, the data source is introduced, the analysis is described, and the resulting distributions and some interesting findings are presented. Secondly, required simulation lengths and times of initial transients are determined. For this purpose, the probabilistic simulation approach and the simulation model are 
[revised manuscript text omitted]
_{\text{water}} = A(T_{\text{water}}) + B(T_{\text{water}})S + C(T_{\text{water}})S^{1.5} + DS^2, \tag{6}$$

where $S$ is the salinity, $T_{\text{water}}$ is the water temperature at the surface, $A$, $B$ and $C$ are polynomial functions of the water temperature and $D$ is a constant. As constant salinity is assumed, the water density is a function of the water temperature. For all wave parameters, three-hour mean values are calculated, as wave conditions stay stationary for a duration of about three hours (GL, 2012). For the speeds and directions of sub- and near-surface currents, measured current values ($v_{\text{m}}$ and $\theta_{\text{m}}$) have to be converted in order to separate sub- and near-surface components. According to, for example, IEC (2009), the following two equations apply for sub- and near-surface currents respectively:

$$v_{\text{SS}}(z) = v_{\text{SS}}(0\,\text{m})\left(\frac{d-z}{d}\right)^{\frac{1}{7}} \quad \text{and} \tag{7}$$

$$v_{\text{NS}}(z) = \begin{cases} v_{\text{NS}}(0\,\text{m})\left(\frac{20\,\text{m}-z}{20\,\text{m}}\right) & \text{for} \quad z <= 0 \\ 0 & \text{for} \quad z > 0. \end{cases} \tag{8}$$

Here, $v_{\text{SS}}(z)$ and $v_{\text{NS}}(z)$ are the sub- and near-surface current speeds at a position $z$ below the water surface, and $d$ is the water depth. For reasons of clarity, the following notation is introduced: $v_{\text{SS}}(z) = v_{\text{SS},z}$. The velocity profiles are shown in Fig. 2. Obviously, the near-surface current does not exist below a reference depth of 20 m. Hence, it is possible to use measurement data of a depth of 20 m (or more) to directly get the sub-surface direction ($\theta_{\text{SS},20} = \theta_{\text{m},20}$) and to calculate the speed, for example for FINO2 ($d = 25\,m$):

$$v_{\text{SS},0} = v_{\text{SS},20}\left(\frac{25\,\text{m} - 20\,\text{m}}{25\,\text{m}}\right)^{-\frac{1}{7}}. \tag{9}$$

For the near-surface current, measurements close to the surface (e.g. $v_{\text{m},2}$) can be used. However, these measurements include sub- and near-surface components, as shown in Fig. 3. Therefore, the sub-surface component at 2 m has to be calculated

[Figure]

**Figure 3.** Vectorial analysis of ocean current components at a depth of 2 m (measured values (m), near- and sub-surface components (NS and SS))

using Eq. (7), and the sub-surface direction is assumed to be constant over depth ($\theta_{\text{SS},20} = \theta_{\text{SS},2} = \theta_{\text{SS},0}$). Then, trigonometrical relationships can be applied to calculate the near-surface current at 2 m:

$$v_{\text{NS},2} = \sqrt{v_{\text{SS},2}^2 + v_{\text{m},2}^2 - 2 v_{\text{SS},2} v_{\text{m},2} \cos(\theta_{\text{m},2} - \theta_{\text{SS},2})} \tag{10}$$

$$\theta_{\text{NS},2} = \theta_{\text{m},2} + \arcsin\left( v_{\text{SS},2} \frac{\sin(\theta_{\text{m},2} - \theta_{\text{SS},2})}{v_{\text{
[revised manuscript text omitted]
 overall probabilistic simulation approach is explained, as it differs from the approach in the standards. Subsequently, the utilised simulation model and the chosen environmental conditions are briefly presented.

**3.1 Probabilistic simulation approach**

For the design of offshore wind turbines, several design load cases (DLC1.1 to 8.3) have to be simulated according to the standards (IEC, 2009). These load cases cover ultimate and fatigue loads during power production, idling and fault conditions, and several special cases like start-up or shut-down. Stochastic inputs for turbulent wind and irregular wind are included. Nevertheless, the DLCs remain quasi deterministic, as environmental conditions like turbulence intensities, wind shear, etc. do not scatter. In order to guarantee safe designs despite the deterministic approach, several ULS load cases, covering extreme environmental conditions (e.g. DLC1.3 for turbulence or DLC1.5 for wind shear), are needed.

In this work, statistically scattering environmental conditions are applied, and therefore, a probabilistic simulation approach is used. This probabilistic approach differs from the deterministic load case based approach. For the probabilistic approach or "real-life" approach, it is not necessary to simulate any load cases of extreme environmental conditions (e.g. DLC1.3 to 1.6), but the use of scattering conditions leads directly to simulations that represent the real lifetime of the turbine (without fault, start-up or other special situations). Hence, simulations (e.g. 10000 simulations) cover a realistic period of power production and idling, leading to about 2.3 months of turbine lifetime (for 10000 simulations). As environmental conditions scatter, effects like high turbulences, extreme wind shear, high waves, small wave periods, and others are covered, and do not have to be considered separately. Load cases are not simulated explicitly, but are cover implicitly by conducting probabilistic simulations.

That is why for FLS, the two approaches do not differ significantly. The "real-life" approach covers DLC 1.2 and 6.4. For ULS, the "real-life" approach covers all power production cases (DLC 1.1-1.6) and DLC 6.1 by applying scattering environmental conditions. As the "real-life" approach cannot simulate 20 years of turbine lifetime (or even a return period of 50 years), a load extrapolation, as required for DLC 1.1, is needed in order to calculate an ULS design. However, this extrapolation is not needed here, as it does not influence the investigated simulation constraints.

As common in academia, only power production and idling is simulated. Fault cases, start-up, etc. is not taken into account due to several reasons. Firstly, at least for the jacket, fault cases are less relevant (Vemula et al., 2010). Secondly, these load cases are very controller and design dependent and need special treatment (e.g. there is no need of removing initial transients for start-up load cases). And thirdly, this work is not intended to calculate exact fatigue damages or ultimate loads for the

whole turbine lifetime, as no turbine design or optimisation is done. The exclusion of some load cases does not affect the recommendations on simulations constraints that are given for power production and idling conditions. As there is no need of exact FLS and ULS lifetime loads in this study, an assessment of the probabilistic approach concerning accordance with the standards is neither conducted nor needed, but would be valuable for further applications of probabilistic approaches.

**3.2 Simulation setup**

As environmental conditions vary for various turbine sites, a data basis being used for the studies of convergence has to be chosen. The basis developed in this work is appropriate, and the FINO3 site is chosen. Some conditions, like air and water density, are kept fixed, as it was shown that their variation is of minor importance (Hübler et al., 2017). It is tried to keep the convergence study as simple as possible, and to focus on the most relevant parameters. Hence, for the probabilistic approach, statistically scattering values according to the determined distributions of wind speed and direction, wave height, direction and period, turbulence intensity, and wind shear exponent are used in all simulations. In addition, the following assumptions are made for all simulations:

- The turbulent wind field is computed according to the Kaimal model and using the software TurbSIM (Jonkman, 2009) with a different wind seed for each simulation.

- Irregular waves are calculated according to the Pierson-Moskowitz spectrum using varying wave seeds for all simulations.

- Soil conditions of the OC3 model (Jonkman and Musial, 2010) are applied.

- The current, second-order and breaking waves, wave spreading effects, marine growth, local vibration effects of braces, joint stiffnesses, and degradation effects are neglected.

The time domain simulations of the convergence study are conducted using the aero-servo-hydro-elastic simulation framework FASTv8 (Jonkman, 2013). A soil model (Häfele et al., 2016) applying linearised soil-structure interaction matrices enhances this code. The NREL 5 MW reference wind turbine (Jonkman et al., 2009) with two different substructures is investigated: Firstly, the OC3 monopile (Jonkman and Musial, 2010) and secondly, the OC4 jacket (Vorpahl et al., 2013). The outcomes of the FAST simulations are, inter alia, time series of forces, moments, and stresses for each element of the substructure.

Since the convergence of fatigue and ultimate loads is investigated in the next step, the calculation concept of these two loads is briefly explained.

For the jacket, the procedure of the fatigue analysis in accordance with DNV-RP-C203 (2010) is the following: For each connection of each joint (K-joints, Y-joint, butt-wlds, etc.), eight hot spot stresses around the circumference of the intersection have to be calculated using the time series. The needed stress concentration factors (SCF) depending on the joint geometry are calculated according Appendix B of DNV-RP-C203 (2010). The fatigue damage is calculated with a fatigue limit of 52.6 MPa at $10^7$ cycles. This corresponds to the DNV-GL S-N curve 90 (for cathodic protection) as used in the original design (Vemula et

al., 2010). For all stresses, a Rainflow counting evaluates the stress cycles. As recommended by the current standards, the conservative damage accumulation according to the Palmgren-Miner rule is assumed using a slope of the S-N curve of three before and five after the fatigue limit for both substructures. The separated fatigue calculation (and summation over all simulations) for each connection of each joint is necessary, as damages in each connection and joint are different for each simulation, and the highest values do not always occur in the same joint (for example due to the probabilistic variation of the wind direction). Finally, the decisive damage for the jacket is the highest accumulated value of all connections of all joints.

For the monopile, the fatigue procedure is similar, but is done according to Eurocode 3, part 1-9 (2010), where a detail of 71 MPa for transverse butt welds and an additional reduction due to the size effect ($t > 25$ mm) is recommended. Differing from the recommendations in Eurocode 3, part 1-9 (2010), the same slopes of the S-N-curves as for the jacket are used.

For the ULS analysis, maximum stresses are decisive and extracted from the time series. For the monopile, Eurocode 3, part 1-6 (2010), is used to analyse the plastic limit state, cyclic plasticity limit state, and buckling limit state (LS1-3). For the jacket, NORSOK N-004 is applied for tubular members and joints which takes combined axial, shear, bending and hydrostatic loadings into account. In both cases, the yield stress is 355 MPa.

Additionally, ultimate limit state proofs for the foundation piles are performed including axial and lateral soil proofs according to GEO2 (DIN 1054, 2010) and a plastic limit state proof (LS1) for the steel pile below mudline. Especially for the monopile, the last proof might be decisive as the bending moment frequently reaches its maximum below mudline. For all ULS proofs, utilisation factors, being the percentage of the maximum loads, are the outcomes.

**3.3 Simulation length**

The simulation length significantly influences the overall computing time of the load assessment. However, there is no conclusive consensus concerning the length needed. Current standards recommend for example 10-minute or one-hour calculations. The offshore oil and gas industry prefers simulation lengths of six hours to cover all low-frequency hydrodynamic effects.

The use of 10-minute simulations can potentially reduce the computing time by a factor of about 36 compared to six-hour simulations. Hence, a study of convergence for bottom fixed offshore wind turbines is conducted here. For floating wind turbines, it is referred to Stewart (2016), who showed that for floating structures all physical effects can be covered with 10-minute simulations.

The presented outcomes of this study focus on the monopile substructure, but a jacket is analysed as well and results (not shown) are generally comparable. For several wind speed bins, 500 simulations with a total length of ten hours are conducted. As the initial transient behaviour is analysed subsequently, a clearly sufficient time, being discarded to exclude the initial transients, of four hours is chosen. Eliminating these four hours of initial transients, the total length of 10 h reduces to a maximum available length (simulation length) of 6 h for the convergence study. In a first step, the convergence of FLS loads is analysed. Afterwards, the ULS case is investigated.

The procedure to calculate the mean fatigue damage for each wind speed bin is the following: From the basis of the 500 ten-hour simulations having different random seeds and varying environmental conditions, 500 cases are selected (with re-

[Figure]

[Figure]

[Figure]

**Figure 10.** Normalised mean fatigue damage (500 simulations) for increasing simulation lengths and different wind speeds.

**Figure 11.** Normalised mean fatigue damage (500 simulations) for increasing simulation lengths and $v_s = 9\text{-}11\,\mathrm{m\,s^{-1}}$. Environmental conditions are kept constant to demonstrate the effect of merging time series more clearly.

placement). For each simulation, the fatigue damage is calculated and weighted with the simulation length. The mean value of all cases is calculated. This procedure is repeated 10 000 times (bootstrapping) to assess the associated uncertainty.

Figure 10 displays the normalised mean fatigue damages for different wind speeds and simulation lengths between ten minutes and six hours. The values are normalised with the six-hour values, and error bars show the $\pm\sigma$ confidence intervals (68 %) that are estimated using a bootstrap procedure with 10 000 resamplings.

It is apparent that due to scattering environmental conditions and the limited number of simulations the uncertainty is relatively high. A detailed investigation of the fatigue load uncertainty, when scattering environmental conditions are applied, is valuable, but out of the scope of this work (cf. Sec. 4). Nevertheless, from Fig. 10 it is apparent that there are no pronounced trends for changing simulation lengths. A slight increase of fatigue loads for higher simulation lengths might be suspected given the fact that such behaviour was observed for floating substructures by Stewart (2016). In order to focus on the simulation length effects, the variation of environmental conditions is neglected in a second step (only varying random seeds). This reduces the uncertainty making it possible to clearly identify a slight increase of FLS loads of about 5 % for higher simulations lengths (see Fig. 11, not merge case). However, as shown by Stewart (2016) for floating substructures, the increasing fatigue loads are not due to any physical effect (all important low-frequency effects of waves are already covered by 10-minute simulations), but can be explained by the effect of unclosed cycles in the Rainflow counting. Cycles that are not completed at the end of the simulation are approximated by counting them as half cycles. The longer the simulation, the less influential is this approximation, as the number of half cycles compared to the number of full cycles reduces. A quite straightforward approach to reduce the problem of half cycles is to merge several shorter simulations (e.g. 10-minute simulations) to a longer one (e.g. six-hour simulation). This means fatigue damages are not calculated for each time series separately, but for longer time series consisting of several shorter ones that are just appended to each other. It is either possible to append different 10-minute time series to each other, or each time series is duplicated and appended several times to itself. If scattering environmental conditions are assumed, in some simulations, fairly different load levels occur. In these cases, load levels of the simulations might not fit,

[Figure]

**Figure 12.** Normalised mean ULS utilisation factor (500 simulations) for increasing simulation lengths and different wind speeds.

[Figure]

**Figure 13.** Normalised mean ULS utilisation factor for increasing simulation lengths (constant overall length of $500 \times 10$ min leading to 500 to 14 simulations) and different wind speeds.

and additional cycles can be introduced by merging different time series, leading to unreasonably increased fatigue damages. Merging each time series with itself, guarantees fitting load levels. On the downside, the computing time of the post-processing is slightly increased. The effect of merging several shorter simulations with itself to generic and repetitive six-hour time series (e.g. each 10-minute time series is duplicated 36 times and is appended to itself to create a six-hour time series) is demonstrated

5 in Fig. 11. It can be seen that the simulation error of about 5 % too low FLS loads for not merged 10-minute simulations can be compensated by merging time series in the post-processing.

For the ULS loads, the calculation procedure is similar. From the basis of the 500 ten-hour simulations, 500 cases are selected (with replacement). The maximum value of all simulations is taken as decisive utilisation factor. This procedure is repeated 10 000 times (bootstrapping) to assess the associated uncertainty.

10 The convergence is shown in Fig. 12. Obviously, ULS loads are higher for longer simulations. Again, this increase is not due to any physical phenomenon, but a result of different overall computing times. Clearly, 500 10-minute simulations should not be compared to 500 six-hour simulations, but to about 14 six-hour simulations (Haid et al., 2013). Therefore, in a second step, the ULS calculation procedure is slightly adapted. Now, 500 cases are only selected for 10-minute simulations. For all other simulations length, the number of cases is reduced to keep the over simulation length constant at 5 000 minutes (i.e. 250 cases

15 for 20-minute simulation, etc.). This comparison is displayed in Fig. 13 and makes clear that ULS loads do not depend on the simulation length but only on the overall computing time. A second fact being visible in Fig. 13 are the higher uncertainties for longer simulation lengths. Since 10-minute simulations lead to a higher number of cases than six-hour simulations for the same total length (i.e. 500 and 14), shorter simulations better cover rare cases, and therefore, scattering environmental conditions leading to less uncertainty.

20 After all, the investigations of this section suggest that simulations of ten minutes length are sufficient independent of the type of load or investigated substructure, or wind speed. At this point, it has to be noted that only two types of substructures are analysed and environmental conditions typical for the North Sea. For significantly different substructures or locations, the validity might be limited. Notwithstanding the above, for ULS loads, the same overall time has to be compared in order to achieve

reliable results. By keeping the simulation length short, more simulations can be conducted in the same overall computing time leading to a better convergence of ULS loads. For FLS loads, simulation errors due to the simulation length can be reduced by merging the time series.

**3.4 Initial transients**

For the analysis of the simulation length, the first four hours of each simulation were discarded to guarantee a steady state operation of the turbine. However, removing four hours of initial transients and only using ten minutes of simulation is computationally very expensive. Therefore, the convergence of FLS and ULS loads with respect to the time of initial transients is analysed. As initial conditions, like an initial rotor speed, influence the initial transient behaviour (Haid et al., 2013), initial rotor speeds and blade pitches depending on the wind speed are set here. These initial conditions are quasi-static states determined using prior simulations.

As the initial transient behaviour is affected by the type of substructure and the load condition, the time that has to be removed is analysed in each wind speed bin for FLS and ULS loads and for both types of substructures separately. Commonly, time series are investigated to estimate times of initial transients (Zwick and Muskulus, 2015). Although this is a straightforward approach, here, it is considered to be not expedient. For a fatigue assessment, the convergence of the fatigue damage has to be analysed, and for the ULS analysis, maximum loads or utilisation factors have be considered.

For each wind speed bin, 10 000 simulations for the monopile and 500 for the jacket were conducted according to the simulation setup in section 3.2. This means: Each simulation has its own random seed for irregular waves and turbulent wind, and in addition, different wind speeds and directions, wave heights, directions and periods, turbulence intensities and wind shear exponents according to the FINO3 data are applied. The high and unequal number of simulations is needed to exclude effects of the number of simulations, mentioned in the previous section and addressed in Sec. 4, as well as possible. For the monopile, each simulation at operating conditions is 900 s long (600 s simulation length plus 300 s of initial transients) and 1800 s at idling conditions. When the turbine is idling, the aerodynamic damping is lower, leading to more pronounced initial transients. For the jacket, all simulations are 720 s long. Using this simulated data basis, it is possible to analyse the effect of different initial simulation times removed on the fatigue damage and utilisation factors in order to determine optima. The analysed simulation length is kept constant at 600 s while the removed length varies between 0 s and 300 s (1200 s for idling; 120 s for the jacket).

Figure 14 displays the convergence of the fatigue damage of the monopile substructure at operating conditions. Here, 300 s or 120 s values are used as a reference, the so called "converged value". The ten-hour simulations in section 3.3 were used determine these values, where the error due to initial transients can be neglected and is much smaller than the error due to the number of simulations. For idling conditions (not shown), the initial transient behaviour takes longer, as the aerodynamic damping is lower. For the same reason, the transients are shorter for higher wind speeds. For the jacket substructure displayed in Fig. 15, the transients decay much faster in all wind speed bins. As jackets are less influenced by wave loads, being not always aligned with the wind, the aerodynamically marginally damped side-to-side modes are less excited, leading to a shorter transient behaviour. This interpretation is supported by the fact that for the jacket, idling conditions, where the hydrodynamic

[Figure]

[Figure]

**Figure 14.** Initial transient behaviour of the operating wind turbine with a monopile substructure for different wind speeds. Percentage difference in the fatigue damage compared to the "converged" value (300 s).

**Figure 15.** Initial transient behaviour of the wind turbine with a jacket substructure for different wind speeds. Percentage difference in the fatigue damage compared to the "converged" value (120 s).

behaviour dominates, have shorter initial transients.

The convergence of ULS utilisation factors for both substructures is shown in Fig. 16 and 17. It becomes apparent that initial transients are short independent of the type of substructure and wind speed. The cycles with high amplitudes occurring at the beginning of each simulation are damped out within a few seconds, and hence, are not influencing the ULS behaviour. More

5    problematic are less damped cycles with smaller amplitudes leading to the previously presented, higher times of initial transients for FLS loads.

The recommended times that should be discarded to exclude initial transients for both substructures, being always a compromise between computing time and accuracy (here, errors below 5 %), are summarised in Table 3. It has to be mentioned that the general validity is limited, as these times of initial transient might vary for example for different aero-elastic codes,

10    numerical solvers, time constants of the aero-elastic models, or substantially different substructures. For example, jackets for 10 MW turbines might behave differently due to larger diameters of legs and braces increasing wave effects. However, for similar applications (e.g. FASTv8, NREL 5 MW turbine, OC3 monopile or OC4 jacket, etc.) that are not rare in academia (e.g. Zwick and Muskulus (2015) or Morató et al. (2017)), the given values represent a well-founded guidance for simulation set-ups. Furthermore, these results shall sensitise the research community to the problem of initial transients especially in case of

15    fatigue. For fatigue, the time of initial transients might be higher than frequently presumed in literature. This is due to weakly damped cycles with small amplitudes that cannot directly be identified when looking at time series.

**4   Benefits and limitations**

[revised manuscript text omitted]

**5 Conclusions**

This work aims to help future simulation work to be more realistic and accurate. In order to achieve this objective, a freely available and comprehensive data basis for scattering environmental conditions was set up. This data basis consists of conditional statistical distribution for many parameters and can be applied without further post-processing. All needed information (statistical distribution and their parameters) is given in the supplementary material. In academia, this data basis enables simulations with probabilistic environmental conditions making them more realistic. For industry purposes, this work might lead to a reconsideration of the current practice. This study shows that the use of deterministic values being either only dependent on the wind speed (e.g. turbulence intensity) or even totally constant (e.g. wind shear) does not represent realistic offshore conditions. However, for a well-founded reconsideration of the current practice, a detailed assessment of probabilistic approaches compared to deterministic load case based ones is needed.

Additionally, scientifically sound recommendations are given for the choice of simulation lengths and times to be removed to exclude initial transients. Simulation lengths of 10 minutes are generally sufficient, and can even help to reduce uncertainties. However, in case of FLS loads, times series should be merged, and for ULS situations, the overall computing time has to be kept constant. Recommendations concerning the initial transients have to be handled with care due to limitations of the general

validity. The values are summarised in Table 3 and can help to improve the accuracy of simulations, and to reduce computing times. It should be noted that a partly significantly longer initial transient behaviour compared to values in literature, being mainly based on educated guesses, was detected.

An enlargement of the current data basis to include additional offshore sites, other types or designs of substructures or investigations for other simulation codes and numerical solver would be definitely valuable to increase the general validity. Furthermore, even for the utilised FAST code, additional investigations concerning the amount of eigenmodes representing the substructure would be beneficial, as a reduction of retained eigenmodes might reduce the time of initial transients.

[revised manuscript text omitted]

---

## Author Response (AR2)

**Answer to: Associate Editor Decision: Publish subject to technical corrections** (25 Sep 2017) by Clemens Hübler

Dear Prof. Muskulus,

thank you for acting as the associated editor for our paper.

The Supplementary Material was modified. In addition to the MATLAB files, we now provide all needed information in ASCII text files. The explanation in the supplement pdf were modified to give explanations regarding the ASCII files.

With best regards,
Clemens Hübler